# SELF-IMPROVING ROBUST PREFERENCE OPTIMIZATION

**Eugene Choi** *
Cohere

**Arash Ahmadian**
Cohere for AI

**Matthieu Geist** †
Cohere

**Olivier Pietquin** †
Cohere

**Mohammad Gheshlaghi Azar** *
Cohere

## ABSTRACT

Online and offline `RLHF` methods, such as `PPO` and `DPO`, have been highly successful in aligning AI with human preferences. Despite their success, however, these methods suffer from fundamental limitations: **(a)** Models trained with `RLHF` can learn from mistakes or negative examples through RL mechanism or contrastive loss during training. However, at inference time, they lack an innate self-improvement mechanism for error corrections. **(b)** The optimal solution of existing methods is highly task-dependent, making it difficult for them to generalize to new tasks. To address these challenges, we propose Self-Improving Robust Preference Optimization (`SRPO`), a practical and mathematically principled offline `RLHF` framework. The key idea behind `SRPO` is to cast the problem of learning from human preferences as a self-improvement process, mathematically formulated as a min-max objective that jointly optimizes a self-improvement policy and a generative policy in an adversarial fashion. Crucially, the solution for this optimization problem is independent of the training task, which makes it robust to its changes. We then show that this objective can be reformulated as a non-adversarial offline loss, which can be efficiently optimized using standard supervised learning techniques at scale. To demonstrate `SRPO` 's effectiveness, we evaluate it using AI Win-Rate (WR) against human (GOLD) completions. When tested on the XSum dataset, `SRPO` outperforms `DPO` by a margin of **15**% after 5 self-revisions, achieving an impressive **90**% WR. Moreover, on the challenging Arena-Hard prompts, `SRPO` outperforms both `DPO` and `IPO` (by **4**% without revision and **6**% after a single revision), reaching a **56**% WR against against `Llama-3.1-8B-Instruct`.

## 1 INTRODUCTION

Reinforcement Learning from Human Feedback (`RLHF`) (Christiano et al., 2017) has quickly become a standard method to align large language models (LLMs). A key advantage of `RLHF` is that it allows the model to learn from its mistakes, allowing it to not only learn from *good* experiences but also from *bad* experiences and mistakes. Despite being able to learn from mistakes, the standard `RLHF`-tuned models are not equipped with an innate mechanism to self-improve and correct their mistakes at the time of inference (Kumar et al., 2024). If a model makes a critical mistake or omits crucial fact during inference, it struggles to recover. Another practical issue that all the prominent `RLHF` methods (offline or online) (Ouyang et al., 2022; Rafailov et al., 2023; Azar et al., 2023; Zhao et al., 2023b; Ahmadian et al., 2024) encounter is that their optimal solution heavily depends on the training task in terms of the distribution used to generate preference data (*behavior policy*) (Munos et al., 2023; Azar et al., 2023). This makes existing `RLHF` methods vulnerable to the cases in which the evaluation distribution of completions is significantly different from that of the behavior policy (Li et al., 2024b; Kirk et al., 2024).

To address these challenges, we introduce an alternative approach for aligning LLMs from human preferences based on more principled and robust foundations. Our goal is to find a solution that is

---

*Corresponding Authors: `{eugene,mohammad}@cohere.com`
†Research was done at Cohere. Now at Earth Species Project.

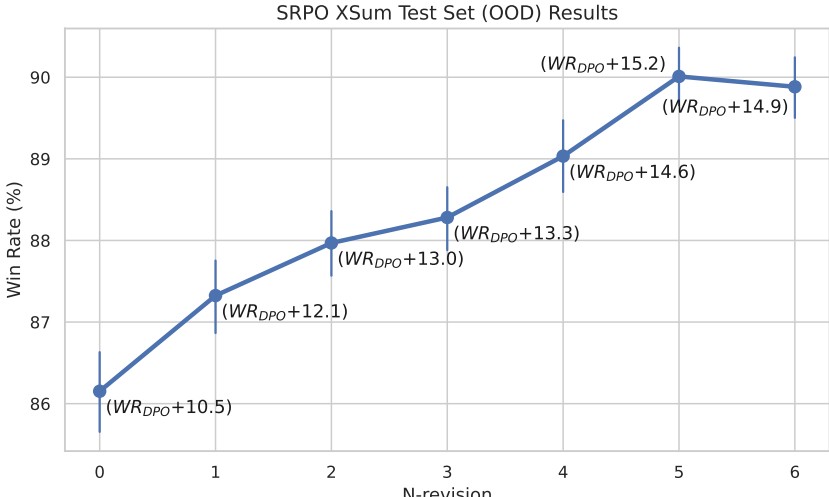

Figure 1: Self-improvement by `SRPO` in terms of win rates against human (*WR*). We demonstrate robustness by training on TL;DR and evaluating on XSum. Gains on Direct Preference Optimization (`DPO`) are reported in text captions.

robust to the changes in the preference dataset, meaning that changes in the distribution from which the completions are sampled do not affect the final outcome of learning significantly. To achieve this goal, we exploit the concept of self-improving (Huang et al., 2022; Bai et al., 2022) language models. By self-improving LLM we refer to a model capable of enhancing its outputs recursively with each inference iteration. Our **S**elf-Improving **R**obust **P**reference **O**ptimization (`SRPO`) consists of two back-to-back optimization processes:

**(Step 1) In-Context Self-Improving Preference Optimization:** The core idea is to learn an in-context self-improving model $\pi_\dagger$:[1] given an *in-context* completion $y$ and a context $x$, the self-improvement model, $\pi_\dagger$, outputs an improved completion $y'$ with probability $\pi_\dagger(y'|y,x)$ from which sampled completions are most preferred to completion $y$ according to the human preference model $p$. As explained later, it turns out that this problem, in its KL-regularized form, can be expressed as a well-defined preference optimization problem and solved analytically. Furthermore, the solution can be estimated through a supervised direct preference optimization scheme similar to the approach used by Rafailov et al. (2023) and Azar et al. (2023).

**(Step 2) Robust Preference Optimization of Generative Model:** The next step is to exploit the self-improvement policy learned in the previous step to learn a generative LLM, $\pi$. The key idea here is that the best generative policy can be identified as a policy that generates completions requiring minimal improvement using the optimal self-improvement policy $\pi_\dagger$ derived in step 1. This goal can be achieved by minimizing the objective of step 1 with respect to the generative policy for *in-context* completions, $y$. Similarly to step 1, this problem, in its KL-regularized form, can also be solved analytically in terms of the optimal improvement policy $\pi_\dagger$ and the optimal generative policy $\pi$. More significantly, we show that the solution for steps 1 and 2 can be estimated jointly through a single supervised direct preference optimization scheme using only a dataset of annotated pair-wise completions. Thus, one can solve both for the self-improvement policy $\pi_\dagger$ and $\pi$ by minimizing the supervised learning objective of `SRPO`. Unlike existing `RLHF` methods, this solution is independent of the behavior policy and is therefore robust to its changes.

As using the self-improvement model in `SRPO` is a significant departure from the existing paradigm for `RLHF`, we provide a high-level motivation for it in Sec. 2. We then formalize our objective for `SRPO` in Sec. 3, allowing for the joint optimization of both $\pi_\dagger$ and $\pi$ by optimizing an adversarial min-max objective. In Sec. 4 we present our main algorithmic/mathematical contribution: we prove that the preference probability $p$ can be expressed in terms of the log-likelihoods of the optimal self-improvement policy $\pi_\dagger^*$ and the log-likelihoods of the optimal robust generative policy $\pi^*$. This

---

[1]From now on, a generative LLM will be considered as equivalent to a distribution or policy $\pi$ from which we can sample completions $y$ with probability $\pi(y|x)$, where $x$ is the context or prompt.

theoretical finding is the key result for `SRPO`: solving this system of equations through least-squares regression provides us with the practical supervised `SRPO` objective that solves for both policy and robust generative policy through a single supervised objective without any need for reward model or online inference. Our key theoretical finding is similar to the main result of `DPO` (Rafailov et al., 2023) in that both express preference probabilities in terms of the optimal policy. However, `DPO` result only holds when preference probabilities conform to Bradly-Terry model (Bradley & Terry, 1952), whereas our key result is general as it holds across all preference models. In Sec. 5 we further illustrate our argument on the robustness of `SRPO` by providing an in-depth analysis of the solution of `SRPO` and other direct preference optimization methods. We also showcase/analyze the robustness of `SRPO` on a simple synthetic example. Finally in Sec. 6 we conduct large-scale experiments on training LLMs with `SRPO` both on in-distribution and OOD summarization tasks and we compare the results with those of standard baselines.

## 2 LEARNING SELF-IMPROVEMENT POLICY THROUGH PREFERENCE OPTIMIZATION

The goal of this section is to motivate why learning self-improvement models through preference optimization can be useful for training a GPT-style language model. One of the main challenges that affect the performance of sequence-to-sequence GPT-style models is that they are hard to recover from failure at the time of inference. This problem, which is known as *exposure bias* in the literature on sequence-to-sequence prediction (Xu et al., 2020; Arora et al., 2023; Bengio et al., 2015), is because at the time of inference, GPT-style models generate tokens step by step based on their own previous outputs, not the ground truth. This can lead to error accumulation: if the model generates an incorrect token early in the sequence (or a misleading token that may lead to an error later down the line), it may propagate errors since its subsequent predictions are based on the earlier incorrect (or misleading) token (Arora et al., 2023).

To address this shortcoming of GPT-style models, one may fine-tune the model such that it can revise its incorrect generations through a self-improvement/self-refinement process (Madaan et al., 2024; Hu et al., 2024; Huang et al., 2022; Bai et al., 2022). The simplest way to train such a self-improvement model is through supervised fine tuning (SFT) (Bai et al., 2022; Wei et al., 2023). To train a self-improvement model through SFT, one first need to build a supervised dataset of $(x, y, y^*)$ in which $y$ is some *initial* completion than can be correct or incorrect and $y^*$ is the corrected version of $y$. The improvement model can then be fine-tuned using SFT to predict $y^*$ from the pair $(x, y)$.

However, training a SFT self-improvement model is a challenging task for the following reasons.

1. Creating a supervised dataset of $(x, y, y^*)$ is resource intensive and hard to scale (Huang et al., 2022). Also, the existing standard SFT datasets are not created in this way.

2. Moreover, the SFT training pipeline for self-improvement can simply overfit to predict the correct answer $y^*$ only from the context $x$ and ignore the initial answer $y$ altogether or over-fit to specific refinement pattern (Li et al., 2024a). Thus it may not learn properly how to improve an incorrect $y$ to the correct $y^*$.

Alternatively, one might ask whether one can generalize the direct preference optimization methods (e.g., Rafailov et al., 2023; Azar et al., 2023) for learning self-improvement models using the standard human preference datasets. To answer this question we first try to answer a more fundamental question:

*What is the best use of human preference data?*

To address this question, we observe that human preferences inherently encode valuable information about the relationship between more-preferred and less-preferred completions. This relational information can be leveraged to refine less-preferred completions, guiding them closer to those that better align with human preferences. In essence, this process involves learning a model of alignment mechanics—principles or "rules" that dictate how completions can be iteratively improved in terms of satisfying human preferences. This approach represents a more natural and intuitive learning task compared to directly predicting the highest-preferred completion, which is the focus of standard `RLHF` methods. Directly learning the optimal completion can be a challenging task, particularly when the completion space encompasses the full complexity of human language. Furthermore, the

highest-preferred answer is unlikely to exist in the dataset, especially when completions are generated by large language models (LLMs) that inherently fall short of human-level quality.

Instead, it is more effective to model the improvement process itself: given a query $x$ and an initial completion $y$, the goal is to predict an improved completion $y'$. This shift in perspective transforms the task into learning how to enhance subpar outputs from an LLM, rather than aiming to directly generate perfect responses. By capturing the underlying principles of human preference, the model can iteratively refine suboptimal completions $y$, progressively steering them towards the ideal outcome $y^*$. This framework not only aligns with the iterative nature of human reasoning but also facilitates continuous self-improvement, making it a robust and scalable approach for enhancing the performance of LLMs. In the following, we show how the self-improvement policy, $\pi_\dagger$, is trained alongside the generative policy, $\pi$, using pair-wise preferences.

## 3 SRPO OBJECTIVE

We start by introducing some notations required to establish our theoretical results. Let $x$ and $y$ denote a context and a completion drawn from the space of all possible contexts $\mathcal{X}$ and all possible completions $\mathcal{Y}$, respectively. The large language model (LLM) is represented by the probability distribution (policy) $\pi$ where $\pi(y|x)$ denotes the probability of completion $y$ given the context $x$. In the remainder of this article, we consider three variants of this base LLM, the trainable model $\pi_{\text{train}}$ (for which we use the shorthand $\pi$), the reference model $\pi_{\text{ref}}$ and the behavior model $\mu$ from which the completions in pairwise preference dataset is sampled.

We also introduce self-improvement $\pi(y'|y, x)$ as a model that using a context $x$ and in-context completion (thought) $y$ aims at improving $y$ to better completion $y'$. Similarly to the base LLM, we can define a reference model $\pi_{\text{ref}}(y'|y, x)$ also for the self-improvement model. Let $\mathcal{D} = \{(x, y, y')\}$ be a dataset of contexts and completions where $y$ and $y'$ are drawn independently of $\mu(\cdot|x)$. We then present every pair $y, y'$ to human annotators who express preferences for one of the completions, denoted as $y_w \succ y_l$ where $y_w$ and $y_l$ denote the preferred and dis-preferred actions amongst $\{y, y'\}$ respectively. We then write the true human preference $p(y' \succ y|x)$ the probability that $y'$ is preferred to $y$ knowing the context $x$. The probability comes from the randomness of the choice of the human we ask for their preference. So $p(y' \succ y|x) = \mathbb{E}_h[\mathbf{1}\{h \text{ prefers } y' \text{ to } y \text{ given } x\}]$, where the expectation is over humans $h$.

Consider a reference policy $\pi_{\text{ref}}$, and a real positive regularization parameter $\beta \in \mathbb{R}_+^*$. Then, we define the Self-Improving Robust Preference Optimization objective (SRPO) for every context $x$ as

$$J^*(x) = \min_\pi \max_{\pi_\dagger} \mathop{\mathbb{E}}_{\substack{y \sim \pi(.|x) \\ y' \sim \pi_\dagger(\cdot|y, x)}} \left[ p(y' \succ y|x) - \beta D_{\text{KL}}(\pi_\dagger || \pi_{\text{ref}}|y, x) + \beta D_{\text{KL}}(\pi || \pi_{\text{ref}}|x) \right], \quad (1)$$

with the KL-regularization terms are defined as: $D_{\text{KL}}(\pi_\dagger || \pi_{\text{ref}}|y, x) = \text{KL}(\pi_\dagger(\cdot|y, x) || \pi_{\text{ref}}(\cdot|y, x))$ and $D_{\text{KL}}(\pi || \pi_{\text{ref}}|x) = \text{KL}(\pi(\cdot|x) || \pi_{\text{ref}}(\cdot|x))$.

In nutshell, this objective aims at **(i)** finding the best self-improvement policy $\pi_\dagger^*$ that improves every $y \sim \pi$ optimally w.r.t. the preference distribution $p$, i.e., the improved policy is most preferred to $y$, while keeping $\pi_\dagger^*$ close to the reference policy $\pi_{\text{ref}}$, **(ii)** minimizing the same objective to find the best (robust) policy $\pi^*$ for which the generated completions can be only minimally improved by the optimal self-improvement model $\pi_\dagger^*$. The min-max nature of this objective makes self-improvement effective for all policies close to $\pi_{\text{ref}}$ as we optimize $\pi_\dagger$ in the worst-case scenario.

## 4 OFFLINE SOLUTION FOR OPTIMIZING SRPO OBJECTIVE

In this section, we show that the min-max objective of Eq. (1) can be transformed to a non-adversarial offline supervised loss. Thus it can be optimized at scale using standard optimization techniques.

### 4.1 MAIN RESULT

The optimization problem of Eq. (1) is a non-trivial optimization problem that often requires solving a two-stage adversarial optimization problem through the game-theoretic approaches, which are often

challenging and difficult to scale up, (see e.g., Munos et al., 2023; Rosset et al., 2024; Calandriello et al., 2024, for how we can use game-theoretic approaches/objectives to train LLMs). Here, inspired by Rafailov et al. (2023); Azar et al. (2023), we aim at casting this complex optimization objective as a standard supervised learning problem that can be solved at scale given an offline pairwise preference dataset. The main theoretical result of our work is then as follows

**Theorem 1.** *Given a context $x$ and the behavior policy $\mu(\cdot|x)$ let $\mu(y|x)$ and scalars $\alpha \in [0,1]$, $\beta > 0$ we have that the solution of the min-max objective of Eq. (1) is obtained by minimizing the following loss*

$$L_\alpha(\pi, \pi_\dagger) = (1-\alpha)L(\pi, \pi_\dagger) + \alpha L_\dagger(\pi_\dagger), \tag{2}$$

*where $L(\pi, \pi_\dagger)$ and $L_\dagger(\pi_\dagger)$ are defined respectively as follows*

$$L_\dagger(\pi_\dagger) = \mathop{\mathbb{E}}_{\substack{y,y' \sim \mu(\cdot|x) \\ x \sim \rho}} \left[ p(y' \succ y|x) - \frac{1}{2} - \beta \left[ \log\left(\frac{\pi_\dagger(y'|y,x)}{\pi_{ref}(y'|y,x)}\right) - \log\left(\frac{\pi_\dagger(y|y,x)}{\pi_{ref}(y|y,x)}\right) \right] \right]^2. \tag{3}$$

*and*

$$L(\pi, \pi_\dagger) = \mathop{\mathbb{E}}_{\substack{y,y' \sim \mu(\cdot|x) \\ x \sim \rho}} \left[ p(y' \succ y|x) - \frac{1}{2} - \frac{\beta}{2} \left[ \log\left(\frac{\pi_\dagger(y'|y,x)}{\pi_{ref}(y'|y,x)}\right) + \log\left(\frac{\pi(y|x)}{\pi_{ref}(y|x)}\right) \right.\right. \tag{4}$$
$$\left.\left. - \left(\log\left(\frac{\pi_\dagger(y|y',x)}{\pi_{ref}(y|y',x)}\right) + \log\left(\frac{\pi(y'|x)}{\pi_{ref}(y'|x)}\right)\right) \right] \right]^2.$$

### 4.2 Proof of Main Result

To prove the main result we first notice that the inner-maximization in the objective function of Eq. (1) is an instantiation of KL-regularized RL (Todorov, 2006). Thus it can be solved in analytical form and its solution is given by

$$\pi_\dagger^*(y'|y,x) = \frac{\exp\left(\frac{p(y' \succ y|x)}{\beta}\right) \pi_{\text{ref}}(y'|y,x)}{Z^*(y,x)}, \tag{5}$$

where $Z^*(y,x)$ is the normalization factor. One can easily show that by plugging $\pi_\dagger^*$ in the objective function of Eq. (1) we obtain:

$$J^*(x) = \min_\pi \mathbb{E}_{y \sim \pi(\cdot|x)} \left[\beta(\log(Z^*(y,x)) + D_{\text{KL}}(\pi||\pi_{\text{ref}}|x))\right]. \tag{6}$$

Now by solving Eq. (5) with respect to $p(y' \succ y|x)$ we obtain

$$p(y' \succ y|x) = \beta(\log(\pi_\dagger^*(y'|y,x)) - \log(\pi_{\text{ref}}(y'|y,x)) + \beta\log(Z^*(y,x))). \tag{7}$$

#### 4.2.1 Optimizing the Self-Improvement Policy $\pi_\dagger$

We now prove that minimizing the objective of Eq. (3) gives us the optimal self-improvement policy (solution to maximization in the objective of Eq. (1)). We first notice that using the convention $p(y \succ y|x) = \frac{1}{2}$ Eq. (7) implies

$$\frac{1}{2} = \beta(\log(\pi_\dagger^*(y|y,x)) - \log(\pi_{\text{ref}}(y|y,x))) + \beta\log(Z^*(y,x)). \tag{8}$$

Now by subtracting Eq. (7) from Eq. (8) we derive

$$p(y' \succ y|x) = \frac{1}{2} + \beta \left[ \log\left(\frac{\pi_\dagger^*(y'|y,x)}{\pi_{\text{ref}}(y'|y,x)}\right) - \log\left(\frac{\pi_\dagger^*(y|y,x)}{\pi_{\text{ref}}(y|y,x)}\right) \right]. \tag{9}$$

This is our first key result that expresses preference $p(y' \succ y|x)$ in terms of the optimal self-improvement policy $\pi_\dagger^*$. So we enforce this equation for all $y$ and $y'$ through minimizing $\ell_2$ loss of Eq. (3) which concludes the proof for the self-improvement policy.

### 4.2.2 JOINT OPTIMIZATION OF THE ROBUST GENERATIVE POLICY $\pi$ AND THE IMPROVEMENT POLICY $\pi_\dagger$

In this section, we show that to optimize the min-max objective of Eq. (1) (i.e., solving for $\pi$ and $\pi_\dagger$) one can optimize the loss of Eq. (4). We start by collecting terms in Eq. (8) which implies $\beta \log(Z^*(y,x)) = \beta(\log(\pi_{\text{ref}}(y|y,x)) - \log(\pi_\dagger^*(y|y,x))) - \frac{1}{2}$. Thus, the objective of Eq. (6) can be expressed in terms of $\log(\pi_\dagger^*(y|y,x))$ (up to an additive and multiplicative constant) as follows:

$$J^*(x) \propto \min_\pi \mathbb{E}_{y \sim \pi(.|x)} \left[ \log \left( \frac{\pi_{\text{ref}}(y|y,x)}{\pi_\dagger^*(y|y,x)} \right) + D_{\text{KL}}(\pi || \pi_{\text{ref}}|x)) \right]. \tag{10}$$

Solving this objective with respect to $\pi$ we obtain:

$$\pi^*(y|x) = \frac{\frac{\pi_{\text{ref}}(y|x)}{\pi_{\text{ref}}(y|y,x)} \pi_\dagger^*(y|y,x)}{Z^*(x)} \tag{11}$$

where $Z^*(x)$ is the normalization factor. Again by taking the logarithm from both side we obtain $\log(\pi^*(y|x)) = \log \left( \frac{\pi_{\text{ref}}(y|x)}{\pi_{\text{ref}}(y|y,x)} \pi_\dagger^*(y|y,x) \right) - \log(Z^*(x))$. Now by collecting terms in Eq. (7) and solving for $\log(\pi_\dagger^*(y'|y,x))$ we obtain

$$\log(\pi_\dagger^*(y'|y,x)) = \frac{p(y' \succ y|x)}{\beta} - \log(Z^*(y,x)) - \log(\pi_{\text{ref}}(y'|y,x)) \tag{12}$$

Now by plugging Eq. (5) into Eq. (11) we deduce $\pi^*(y|x) = \frac{\exp(-\log(Z^*(y,x)))\pi_{\text{ref}}(y|x)}{Z^*(x)}$. Solving this equation with respect to $\log(Z^*(y,x))$ implies

$$\log(Z^*(y,x)) = \log(\pi_{\text{ref}}(y|x)) - \log(\pi^*(y|x)) - \log(Z^*(x)). \tag{13}$$

Combining Eq. (12) and Eq. (13) for any $y$ and $y'$ we have $\frac{p(y' \succ y|x)}{\beta} - \log \left( \frac{\pi_\dagger^*(y'|y,x)}{\pi_{\text{ref}}(y'|y,x)} \right) = \log \left( \frac{\pi_{\text{ref}}(y|x)}{\pi^*(y|x)} \right) - \log(Z^*(x))$, and also $\frac{p(y \succ y'|x)}{\beta} - \log \left( \frac{\pi_\dagger^*(y|y',x)}{\pi_{\text{ref}}(y|y',x)} \right) = \log \left( \frac{\pi_{\text{ref}}(y'|x)}{\pi^*(y'|x)} \right) - \log(Z^*(x))$. Subtracting these two equations and collecting terms leads to our *key result* in which we express the preference $p$ in terms of the self-improvement policy $\pi_\dagger^*$ and the robust policy $\pi^*$.

$$p(y' \succ y|x) = \frac{1}{2} + \frac{\beta}{2} \left[ \log \left( \frac{\pi_\dagger^*(y'|y,x)}{\pi_{\text{ref}}(y'|y,x)} \right) + \log \left( \frac{\pi^*(y|x)}{\pi_{\text{ref}}(y|x)} \right) \right.$$

$$\left. - \left( \log \left( \frac{\pi_\dagger^*(y|y',x)}{\pi_{\text{ref}}(y|y',x)} \right) + \log \left( \frac{\pi^*(y'|x)}{\pi_{\text{ref}}(y'|x)} \right) \right) \right]. \tag{14}$$

**Remark 2.** *One may notice the similarity of this result and Eq. 6 of* DPO *paper (Rafailov et al., 2023). Both results express $p(y' \succ y|x)$ in terms of the optimal policy $\pi^*$. However, the result of* DPO *only holds under the assumption that $p$ conforms to the Bradly-Terry model, whereas our result is general and holds for all $p$.*

To optimize for $\pi$ and $\pi_\dagger$ using Eq. (14) we enforce this equation for all $y$ and $y'$ through minimizing the expected $\ell_2$ loss of Eq. (4) which concludes the proof on joint optimization of the generative policy $\pi$ and the self-improvement policy through Eq. (4).

### 4.2.3 PUTTING ALL TOGETHER IN A SINGLE COMBINATION LOSS

We observe that the losses defined in Eq. (4) and Eq. (3) are inherently aligned, as both aim to optimize the same overall objective given in Eq. (1). This alignment allows us to construct a combined loss function for SRPO by using a convex combination of these two losses. Specifically, for any $\alpha \in [0, 1]$, minimizing the convex combination of the losses in Eq. (4) and Eq. (3) is equivalent to directly optimizing the objective in Eq. (1). Mathematically, the full loss of SRPO can be expressed as:

$$\mathcal{L}_{\text{SRPO}} = \alpha \cdot \mathcal{L}_{\text{L2}} + (1 - \alpha) \cdot \mathcal{L}_{\text{self-improve}},$$

where $\mathcal{L}_{\text{L2}}$ corresponds to the loss in Eq. (4) and $\mathcal{L}_{\text{self-improve}}$ corresponds to the loss in Eq. (3). By minimizing this combined loss, we ensure that the optimization process aligns with the original objective in Eq. (1) regardless of the value of $\alpha$.

This formulation not only unifies the two loss functions within a single framework but also provides flexibility in adjusting their relative contributions during training. This completes the proof.

### 4.3 DERIVING THE SAMPLE LOSS FOR SRPO

Calculating/optimizing the expected loss of Eq. (17) is often not practical as we often have no direct access to the behavior policy $\mu$. Moreover, the space of completions $y$ is often very large (even infinite). So calculating the full expectation is often impractical. Instead one can estimate the expected loss $L_\alpha$ of Eq. (17) using a preference dataset $\mathcal{D}$ with a sample loss $\widehat{L}_\alpha$ and optimize this loss instead. We now focus on deriving the sample loss $\widehat{L}_\alpha$ for SRPO. Using the standard properties of $\ell_2$-norm to replace $p(y' \succ y|x)$ with $\mathbf{1}(y' \succ y|x)$, as $p(y' \succ y|x) = \mathbb{E}[\mathbf{1}(y' \succ y|x)]$, in the objective of Eq. (4) allows us to derive the following sample loss for the improvement model:

$$\widehat{L}_\dagger(\pi_\dagger) = \mathbb{E}_{(y_l, y_w, x) \sim \mathcal{D}} \left[ \frac{1}{2} - \beta \left[ \log \left( \frac{\pi_\dagger(y_w|y_l, x)}{\pi_{\text{ref}}(y_w|y_l, x)} \right) - \log \left( \frac{\pi_\dagger(y_l|y_l, x)}{\pi_{\text{ref}}(y_l|y_l, x)} \right) \right] \right]^2 \tag{15}$$
$$+ \mathbb{E}_{(y_l, y_w, x) \sim \mathcal{D}} \left[ \frac{1}{2} - \beta \left[ \log \left( \frac{\pi_\dagger(y_w|y_w, x)}{\pi_{\text{ref}}(y_w|y_w, x)} \right) - \log \left( \frac{\pi_\dagger(y_l|y_w, x)}{\pi_{\text{ref}}(y_l|y_w, x)} \right) \right] \right]^2 .$$

Using the standard properties of $\ell_2$-norm to replace $p(y' \succ y|x)$ with $\mathbf{1}(y' \succ y|x)$, as $p(y' \succ y|x) = \mathbb{E}[\mathbf{1}(y' \succ y|x)]$, in the loss of Eq. (4) allows us to derive the following sample loss for both the generative policy and the improvement model:

$$\widehat{L}(\pi, \pi_\dagger) = \mathbb{E}_{(y_l, y_w, x) \sim \mathcal{D}} \left[ \beta \left[ \log \left( \frac{\pi_\dagger(y_w|y_l, x)}{\pi_{\text{ref}}(y_w|y_l, x)} \right) + \log \left( \frac{\pi(y_w|x)}{\pi_{\text{ref}}(y_w|x)} \right) \right. \right. \tag{16}$$
$$\left. \left. - \left( \log \left( \frac{\pi_\dagger(y_l|y_w, x)}{\pi_{\text{ref}}(y_l|y_w, x)} \right) + \log \left( \frac{\pi(y_l|x)}{\pi_{\text{ref}}(y_l|x)} \right) \right) \right] - 1 \right]^2 .$$

looseness=-1 Furthermore, one can use a single LLM (denoted by $\pi$) to represent both $\pi$ and $\pi_\dagger$ by exploiting the in-context learning power of LLMs (Brown et al., 2020) such that $\pi_\dagger(y'|y, x) = \pi(y'|y, x)$. So for we define the full sample loss of SRPO as the convex combination of Eq. (16) and Eq. (15):

$$\widehat{L}_\alpha(\pi) = (1 - \alpha)\widehat{L}(\pi, \pi_\dagger = \pi) + \alpha \widehat{L}_\dagger(\pi_\dagger = \pi). \tag{17}$$

The following pseudo-code can be used to train the LLM policy using SRPO objective:

---

**Algorithm 1** Sampled SRPO

---

**Require:** Dataset $\mathcal{D}$ of prompts, preferred and dis-preferred generations $x$, $y_w$ and $y_l$, respectively. A reference policy $\pi_{\text{ref}}$ and a training policy $\pi_\theta$, regularization coefficient $\beta$ and combination coefficient $\alpha$.
1: Initialize $\pi_\theta = \pi_{\text{ref}}$
2: **while** true **do**
3:     Sample a minibatch $B \in \mathcal{D}$
4:     Estimate $\nabla_\theta \widehat{L}_\alpha(\pi_\theta)$ from Eq. (17) using minibatch $B$ as the dataset
5:     Update $\pi_\theta$ using $\nabla_\theta \widehat{L}_\alpha(\pi_\theta)$ using a standard optimizer
6: **end while**
7: **return** $\pi_\theta$

---

## 5 ROBUSTNESS OF SRPO

We provide a comparison between SRPO and prior work on direct preference optimization in terms of their robustness to the behavior policy $\mu$. In particular, we consider as a point of reference DPO (PPO[2]) and IPO for which we have a good understanding of the underlying mathematical foundation.

In the case of both IPO and DPO the analytical solution is already well-established and analyzed for both algorithms (Azar et al., 2023; Rafailov et al., 2023; Tang et al., 2024). In particular, the optimal

---

[2]As it is shown by Azar et al. (2023) the optimal solutions of DPO and PPO are identical. So in the remainder of this section we focus on DPO.

solution for both `IPO` and `DPO` can be expressed explicitly in terms of the soft-max of the expected preference as follows (Azar et al., 2023):

$$\pi^*(y|x) = \frac{\exp\left(\beta^{-1}\mathbb{E}_{y'\sim\mu(\cdot|x)}[\Psi(p(y \succ y'|x))]\right)\pi_{\text{ref}}(y|x)}{Z^*(x)}, \tag{18}$$

with the choice of $\Psi = I(\cdot)$ and $\Psi = \sigma^{-1}(\cdot)$ for `IPO` and `DPO`, respectively, where $\sigma^{-1}$ denotes the inverse-sigmoid (logit function). Thus, based on Eq. (18), we can see that the solution for both `IPO` and `DPO` has strong dependency on $\mu$ in the form of expected preference under the distribution $\mu$. Thus it may not be robust to changes in $\mu$. This dependency on $\mu$ can be especially problematic when we evaluate the model on out-of-distribution tasks where the desired behavior is very different from $\mu$ and the expected preference under the distribution $\mu$ is not a good measure of performance. `SRPO` solution on the other hand has no dependency on the behavior policy $\mu$: from Eq. (5) we observe that the optimal self-improvement policy $\pi_{\dagger}^*$ is independent of $\mu$ and, unlike `DPO` and `IPO` cases, is expressed in terms of softmax of $p(y' \succ y|x)$ for any pair of completions $(y, y')$. Also the 0-revision policy $\pi^*$ is also completely independent of $\mu$ as it is evident from Eq. (11) (i.e., it is proportional to $\pi^*(y|y, x)$ which itself is independent of $\mu$). Thus, from a mathematical point of view, `SRPO` provides a robust solution for the problem of direct preference optimization that does not depend on the behavior policy $\mu$. To illustrate the differences between `SRPO` and `DPO`/`IPO` regarding robustness to $\mu$, we consider a simple bandit example. For simplicity, we assume there is no context $x$. Consider the simple case where we have 3 completions $y_0$, $y_1$, and $y_2$, for which the preference model is $P = \begin{pmatrix} 0.5 & 0.99 & 0.3 \\ 0.01 & 0.5 & 0.25 \\ 0.7 & 0.75 & 0.5 \end{pmatrix}$.

To test this hypothesis we consider two synthetic dataset of actions generated from distributions $\mu_0$ and $\mu_1$: We set $\mu_0$ to be a uniform behavior policy ($\mu_0(y_0) = \mu_0(y_1) = \mu_0(y_2) = 1/3$) and $\mu_1$ skewed towards $y_1$ ($\mu_1(y_1) = 0.7, \mu_1(y_0) = \mu_1(y_2) = 0.15$). We then generate a dataset of $10,000$ pairs from $\mu_0$ and $\mu_1$ and rate them according to the preference model $p$ (for any pair $(y, y')$ we assign the preference by sampling from $p(y \succ y')$, that is $y$ is preferred to $y'$ with probability $p(y \succ y')$). This provides us with two dataset of rated completions $\mathcal{D}_0$ and $\mathcal{D}_1$ for $\mu_0$ and $\mu_1$. We then use these two datasets to train the policy $\pi$ using `SRPO`, `DPO` and `IPO` using a simple Adam optimizer. In the case of `IPO` and `DPO` we optimize only the 0-revision policy $\pi(y)$ where as for `SRPO` we also optimize the self-improvement policy $\pi(y|y')$ as well. We set the regularization constant $\beta$ for all methods to 1. We consider a uniform distribution $\pi_{\text{ref}}(y) = 1/3$ for all algorithms and all $y$s. In the case of `SRPO` we set the self-improvement reference policy $\pi_{\text{ref}}(y|y') = 1/3$ for all $y$ and $y'$. Also for `SRPO` we set the combination coefficient $\alpha = 0$ for simplicity.

We observe that in the case of using uniform $\mu_0$ as a behavior policy, all methods perform as expected, converging to solutions where $y_2$ dominates $y_1$ and $y_0$ (Fig. 2a). However, when the behavior policy $\mu_1$ is skewed toward $y_1$, both `DPO` and `IPO` converge to a solution where $y_0$ dominates $y_1$ and $y_2$, while the policy of `SRPO` remains unchanged (Fig.2b). Notably, the `SRPO` policy differs slightly between the two cases, which is expected given the finite data setting, where the sampling distribution influences the empirical preference model." However, when we use the behavior policy $\mu_1$, which is skewed towards $y_1$, both `DPO` and `IPO` converge to a solution where $y_0$ dominates $y_1$ and $y_2$, while the policy of `SRPO` remains unchanged (Fig.2b). Notably, the `SRPO` policy differs slightly between the two cases, which is expected given the finite data setting, where the sampling distribution influences the empirical preference model."

to a solution in which $y_0$ dominates $y_1$ and $y_2$, while the policy of `SRPO` remains intact (Fig. 2b). Notice that the `SRPO` policy is slightly different in both cases. This is to be expected, we are in a finite data setting, and the sampling distribution will have some influence on the empirical preference model.

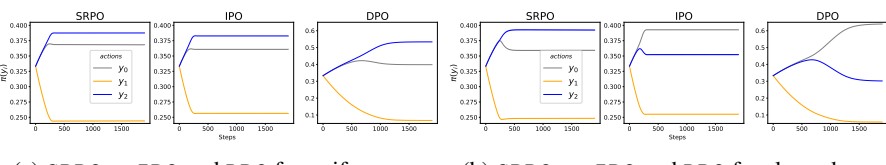

(a) `SRPO` vs `IPO` and `DPO` for uniform $\mu_0$.    (b) `SRPO` vs. `IPO` and `DPO` for skewed $\mu_1$.

Figure 2: Learned action probabilities for the synthetic example. `SRPO` always chooses the correct arm regardless of skew in $\mu$, while both `IPO` and `DPO` are effected by the skew (Fig. (2b)).

## 6 EXPERIMENTS

**Setup.** In our experiments, we consider the offline direct preference optimization setup to learn from human preferences (Rafailov et al., 2023). In the offline setting, the goal is to train the LLM policy directly from a dataset $\mathcal{D}$ of pairwise completions $(y_l, y_w)$ sampled from a behavior policy $\mu$ and annotated by human raters without using a reward model or online inference/RL. We empirically test the effectiveness of SRPO against two prominent offline preference learning methods, DPO (Rafailov et al., 2023) and IPO (Azar et al., 2023) as baselines. We make this choice since both these baselines, have been widely used in solving different language tasks (Tunstall et al., 2023; Wallace et al., 2023; Yuan et al., 2024a; Pang et al., 2024a; Lin et al., 2023).

**Implementation details.** SRPO trains simultaneously both the standard generative policy $\pi$ and the self-improvement policy $\pi_\dagger$ used for revising the completions of models through a single optimization process. As explained earlier we only use a single LLM to represent both $\pi$ and $\pi_\dagger$ (denoted simply by $\pi$). To get the best completions from SRPO we first generate completions in the 0-revision (0-rev.) model and then we improve these completions with the self-improvement model. We call the revised outputs 1-revision (1-rev.) completions. We can iterate on the improvement process $N$ times to get $N$-revision ($N$-rev.) completions. We report results from 0-rev. to 5-rev. cases. For IPO and DPO we also report results on 0-rev. and 1-rev. For revising the completions we use IPO and DPO in in-context learning mode with the 0-rev. completions used as contexts. In the case of DPO we use the same loss and hyper-parameters used by (Rafailov et al., 2023). For IPO since the original paper hasn't provided the hyper-parameters we used a set of hyper-parameters (i.e., learning rate and regularization constant $\beta$) from the range of hyper-parameters that was working. Furthermore we noticed that the performance of IPO was not affected significantly by the choice of these hyper-parameters. So no significant gain is expected by hyper-parameter tuning.

**Datasets.** We use the Reddit TL;DR Summarization dataset (Stiennon et al., 2020) as the main dataset for our experiments[3]. For training, there are 116k human-written instruction following examples with reference completions (SFT split) while there are 93k human-annotated preference pairs (Preference split). We also use the XSum dataset test split[4] (Narayan et al., 2018), which contains 11.5k total test examples to measure Out-of-Distribution (OOD) generalization.

**Model Setup.** We use LLaMA-7B as base model (Touvron et al., 2023) and a single $8 \times$ NVIDIA H100 node to conduct all LLaMA-based experiments. We first SFT the model on the SFT split of the TL;DR dataset, before preference training and use the same $\pi_{\text{ref}}$ for all preference training experiment. Below are details on the recipe for the SFT and preference training stages.

**Supervised-fine Tuning.** In the SFT stage, we train for 2 epochs, using the AdamW optimizer (Loshchilov & Hutter, 2019), with $\beta_1 = 0.9$ and $\beta_2 = 0.999$, and 0.1 weight-decay. We use a cosine decay learning rate (Loshchilov & Hutter, 2017) with a peak value of $2 \times 10^{-5}$ and 3% of all steps being warm-up steps. We use an effective batch-size of 64.

**Preference Training.** We use our SFT model as $\pi_{\text{ref}}$ and we initialize $\pi$ with $\pi_{\text{ref}}$. All models were trained for 5 epochs on the TL;DR preference split using the same optimization setting of the AdamW optimizer as in the SFT stage with 150 warmup steps, and an effective batch-size of 128. To fine-tune the models, we use the default PEFT settings in the TRL library[5], using LoRA (Hu et al., 2022) with a rank of 16 and an alpha of 32. For SRPO and IPO, we used $\beta = 0.01$ with a learning rate of $2 \times 10^{-6}$. For DPO following Rafailov et al. (2023), we used the common $\beta = 0.1$ with a learning rate of $1 \times 10^{-6}$ and a constant learning rate schedule.

**Evaluation.** We use win rates as computed by gpt-4-0613 (OpenAI, 2023) using the Alpacafarm framework (Dubois et al., 2024), as the main means for evaluation (See Sec. C.3 for more details). We measure performance on both in-distribution and OOD examples at test time in the following manner: For the former, we compute win rates against gold reference completions from the test set of the TL;DR SFT split. For the latter, we measure win rates against gold completions from the test set of the XSum dataset. In both settings, we use the first 1,024 samples from each of the test sets. To estimate the win rate more accurately with confidence intervals, we bootstrap 20 times with replacement from the 1,024 samples, each time using a sample size of 512. To sample from the

---

[3] https://github.com/openai/summarize-from-feedback
[4] https://huggingface.co/datasets/csebuetnlp/xlsum
[5] https://github.com/huggingface/trl

self-improvement policy, we first sample $y$ from $\pi(\cdot|x)$. Then, using the same policy, we condition on $y$ to sample from the self-improvement policy, that is $y' \sim \pi(\cdot|y, x)$. We refer to generations from $\pi(\cdot|x)$ as 0-rev. (0-rev.) and generations from $y' \sim \pi(\cdot|y, x)$ as 1-rev. (1-rev.) (Bai et al., 2022). For $N$-revision, we apply the same procedure, conditioning on the sample from the $N$-1$^{\text{th}}$-rev.

**TL;DR Results.** We test our models on the test set split of the TL;DR dataset in Fig 3 (left panel). For every model, we generate 0-rev. and then use these generations to revise our completions recursively from 1-rev. to 5-rev. using the self-improvement model, and measure the models' win rate against the human-written gold reference summaries.

We observe that in the case of in-distribution TL;DR `SRPO` 4-rev. generates high-quality summaries with the highest win rate against the gold summaries, compared to the win-rates of of the baseline methods, as well as other variants of `SRPO` 0-rev. Furthermore, we observe that `SRPO` self-improvement process manages to consistently improve upon `SRPO` 0-rev. . However, `DPO` and `IPO` fail to generate an improved sample through the self-improvement step.

**Out-of-distribution (OOD) Results.** To assess robustness in an OOD setting, we test `SRPO` models trained with TL;DR preference dataset on the XSum test split in Fig. 3 (right panel) (Narayan et al., 2018). As in the TL;DR case, we observe that self-improvement is effective in improving the performance of `SRPO` as `SRPO` 5-rev. generates the highest win rate against the gold summaries, compared to all revisions of the baseline methods, as well as prior revision of `SRPO`. We also observe that the gap in performance between `SRPO` and the baselines is significantly higher in OOD case.

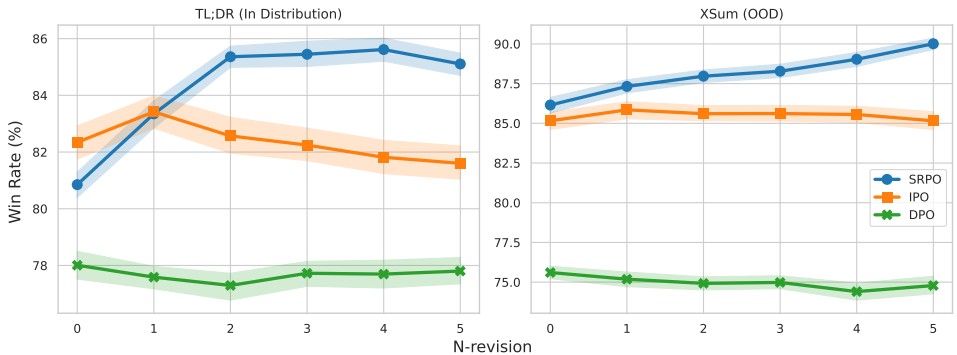

Figure 3: We present the win rates of `SRPO`, `IPO`, and `DPO` against human-written summaries (GOLD) as a function of $N$-revisions for both in-distribution (TL;DR) and out-of-distribution (XSum) settings. The curves represent the mean win rates, with shaded areas indicating the st.dev. across 20 bootstrap evaluations. Notably, `DPO` and `IPO` show no improvements in their generations, whereas `SRPO` shows significant improvements with each iteration.

# 7 DISCUSSION AND LIMITATIONS

In this paper we have developed Self-Improving Robust Preference Optimization (`SRPO`), a robust offline approach for learning from human preferences. We have proven mathematically and empirically, that unlike other offline methods like `DPO` and `IPO`, the solution of `SRPO` is completely independent of the behavior policy $\mu$ and thus `SRPO` is completely robust to changes in $\mu$.

**Summary of results.** We have tested `SRPO` on standard summarization tasks both on in-distribution and out-of-distribution (OOD) regimes. We have observed that in the OOD case `SRPO` outperforms both `IPO` and particularly the celebrated `DPO` by a clear margin in terms of win-rate against gold completions, while in the in-distribution case there is less difference between `SRPO` and the baselines. This is an expected behavior since in-distribution case the robustness aspect of the algorithm matters less. We have observed that although $0$-revision generation of `SRPO` performs well, we have observed a boost across the board by revising the generation through the self-improvement model.

**Future work and Limitations.** In our work we used standard and relatively simple language tasks. In the future we would like to apply `SRPO` to more challenging multi-task benchmarks in which the existing `RLHF` methods often specialize to a specific set of tasks more represented in the dataset, whereas `SRPO` should be more resilient due to its robustness to behavior policy $\mu$.

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

## A    RELATED WORKS

Our work lies in offline preference optimization, a vivid area of research since the introduction of DPO (Rafailov et al., 2023). Some of the core concepts of this research topic was generalized and formalized by Azar et al. (2023). In particular they characterized the underlying optimal solution for a generic preference optimization objective and introduced IPO for addressing some of the related shortcomings of DPO. SLiC-HF (Zhao et al., 2023a) was introduced around the same time, from a less RL-centric point of view. All these approaches have been abstracted later by Tang et al. (2024), the general recipe being to build a contrastive loss function from a convex classification function and to make use of the analytical solution of the RL problem to learn directly the policy. A common underlying assumption is that the related RL problem is KL-regularized. This has been generalized to more general $f$-divergences by Wang et al. (2023). These are just a few among many works on direct alignment (Tajwar et al., 2024; Xu et al., 2024; Guo et al., 2024; Yuan et al., 2024b; Song et al., 2024). However, they all share the fact of not considering self-improvement policies, contrary to SRPO. Concurrent to this work Kumar et al. (2024) use online RLHF for learning self-correcting model. However unlike our method which aims at maximizing the preference of improved completion with respect to the in-context completion their work focuses on maximizing the reward of final answer. The issue with this approach is that the model can simply ignore the in-context completion and try to optimize the final completion without paying attention to in-context completions.

Offline preference optimization was introduced as an alternative to more classic RLHF approaches, such as PPO (Schulman et al., 2017; Ouyang et al., 2022) or more generally policy-gradient-based approaches (Roit et al., 2023; Ahmadian et al., 2024). These methods require training a *reward model* on a preference dataset, usually with a Bradley-Terry model (Bradley & Terry, 1952). The reward model is then used to fine-tune the LLM via online RL, requiring many generations from the model. This reward model shares the common issue of DPO and other direct preference alignment methods, it is dependent on the sampling distribution $\mu$ used for constructing the preference dataset, contrary to SRPO. Moreover, classic RLHF is online, while SRPO is offline and thus more easily scalable.

Some similarities also exist between SRPO and Nash-MD (Munos et al., 2023). Indeed, if in 1 we replace the self-improvement policy $\pi_\dagger(\cdot|y, x)$ by a classic policy $\pi(\cdot|x)$, then we obtain the saddle-point optimization problem that Nash-MD solves. However, considering a self-improvement policy is a core contribution of our work, and it is not anecdotal. From a technical viewpoint, this is critical for simplifying the minimax problem of Eq. (1) and obtaining a simple offline optimization problem. NashMD on the other hand adapts algorithms from the game-theory literature and can only be solved online with all the stability issues of online methods and large inference costs. From practical point of view self-improvement provides a boost in performance by refining the original generations of LLM. The feature that Nash-MD is missing. Finally, even though the Nash equilibrium of Nash-MD does not depend on the sampling distribution $\mu$, it relies on a learned reward function, with the possible associated caveats mentioned earlier, which is not the case of SRPO.

Our work is also obviously related to the concept of chain of thoughts (Wei et al., 2023; Yao et al., 2024), self-improvement (Huang et al., 2022) and self-refining LLMs (Madaan et al., 2024). However, it is very often used as a way of prompting a model to obtain better results, and less often as a component of a learning paradigm (Liu et al., 2023; Huang et al., 2022). A notable exception is the recent work of (Pang et al., 2024b) that generalizes DPO to incorporate chain of thoughts. However the main focus of this work is mostly on improving the chain of thoughts reasoning and not on self-improvement. To our best knowledge, we propose the first approach that combines training self-improvement LLMs and offline preference optimization through a single supervised objective, moreover in a theoretically grounded manner and showing the robustness to $\mu$.

## B    ABLATION: THE EFFECT OF COMBINATION COEFFICIENT $\alpha$ ON SRPO PERFORMANCE

SRPO loss of Eq. (17) is a convex combination of two losses $\widehat{L}$ and $\widehat{L}_\dagger$ via the combination coefficient $\alpha$. To understand how both terms affects the loss we plot the win rates both in in-distribution case and OOD case as a function of $\alpha$ in Fig. 4. We observe that the term that contributes most to the performance of SRPO is $\widehat{L}_\dagger$ as in the case of $\alpha = 1$ when we only use the loss for improvement model $\widehat{L}_\dagger$ we almost match the best performance. On the other hand using only $\widehat{L}$ (i.e., $\alpha = 0$) is not

enough to achieve top performance. We also observe combining both losses seems to provide some boost in performance especially in OOD case.

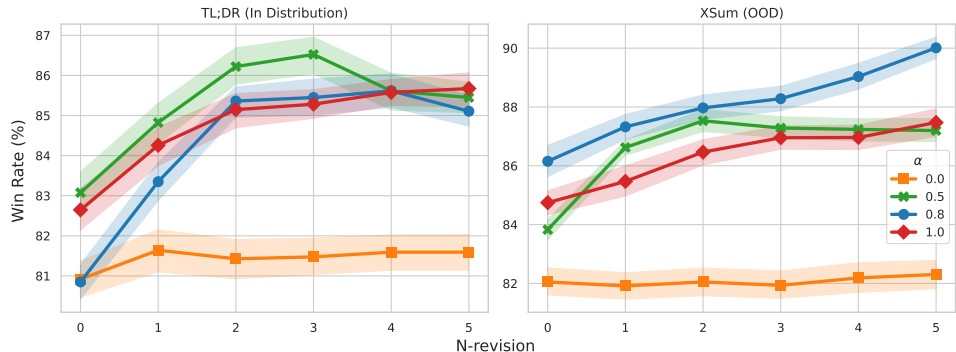

Figure 4: We present the win rates of SRPO against human-written summaries (GOLD) as a function of $N$-revision iterations at different $\alpha$ values. We report their mean (curve) $\pm$ st.dev. (shaded area), across 20 bootstrap evaluations, as described in the **Evaluation** section. We observe that SRPO achieves meaningful iterative improvements capability as the value of $\alpha$ increases.

## C    EXPERIMENTAL DETAILS

We provide the prompt templates used for training and evaluations in section 6.

### C.1    PROMPT TEMPLATES

#### C.1.1    TL;DR

**0-revision:**

```
Below is a reddit POST and the corresponding SUBREDDIT and TITLE.
Write a both precise and concise summary of the contents of the
POST.

SUBREDDIT: ${subreddit}
TITLE: ${title}
POST: ${post}
TL;DR:
```

**N-revision:**

```
Below is a reddit POST and the corresponding SUBREDDIT, TITLE, and
 an EXAMPLE SUMMARY. Write a both precise and concise summary of
the contents of the POST.

SUBREDDIT: ${subreddit}
TITLE: ${title}
POST: ${post}
EXAMPLE SUMMARY: ${(N-1)th_example_summary}
TL;DR:
```

#### C.1.2    XSUM

**0-revision:**

```
Below is a news ARTICLE and the corresponding ID and TITLE. Write
a both precise and concise summary of the contents of the ARTICLE.
```

```
ID: ${id}
TITLE: ${title}
ARTICLE: ${article}
TL;DR:
```

**N-revision:**

```
Below is a news ARTICLE and the corresponding ID, TITLE, and an
EXAMPLE SUMMARY. Write a both precise and concise summary of the
contents of the ARTICLE.

ID: ${id}
TITLE: ${title}
ARTICLE: ${article}
EXAMPLE SUMMARY: ${(N-1)th_example_summary}
TL;DR:
```

## C.2 Example Summaries

### C.2.1 TL;DR

| | |
|---|---|
| Post | *I have a horrible caffeine addiction, and I don't like sacrificing any of my daily calories for coffee. I used to drink 5-6 Diet Dr. Peppers a day, but I have switched to almost exclusively drinking only water most days. I do have a Diet Dr. Pepper occasionally, and I know that technically it's okay to probably have one a day, but I honestly just feel better all around when I drink only water. But if I don't have some caffeine in the morning, I will be writhing in the floor in pain with a skull-pounding headache by mid-day. I know. Ridiculous, right?*

*So, I went to the vitamin aisle the other day in search of a supplement with a little caffeine and the only thing I found was One A Day Adult Vitacraves Gummies with Energy Support. Here's the website with [nutritional info] One serving is 20 calories and has 90 mg of caffeine. There's also some vitamin B6, B12, and folic acid in there that supposedly helps turn food into energy. I have no idea if that's true. I've been taking them a couple days, and so far I do feel more energized and I haven't had any headaches. The issue is that one bottle is around $9 and there are only 60 in a bottle. You have to take 4 (4!) a day, which is very intriguing considering the brand name and all. So it'll cost me around $20 a month. Which is admittedly cheaper than what I used to spend on Diet Dr. Peppers, but still. I was just curious if anyone else had a better option for caffeine in a supplement or vitamin?* |
| GOLD | The vitamin with caffeine I'm taking is on the expensive side, and I'm a self-proclaimed tightwad. Anyone have better options? |
| `SRPO` 1-rev. | I have a caffeine addiction and need a little bit of caffeine in my day. I'm taking One A Day Adult Vitacraves Gummies with Energy Support, but it's expensive ($20/month). Any other suggestions for a caffeine supplement or vitamin? |
| `SRPO` 0-rev. | I have a terrible caffeine addiction and need a little caffeine in my day. I found a vitamin with caffeine in it, but it's pretty expensive. Any other suggestions for a caffeine supplement or vitamin? |
| `IPO` 1-rev. | I have a caffeine addiction and need some in my system every day to function, but I don't like sacrificing calories for coffee. I've found a supplement with caffeine, but it's expensive. Anyone have a better option? |
| `IPO` 0-rev. | I have a caffeine addiction and need some in my system every day to function, but I don't like sacrificing calories for coffee. I've found a supplement with caffeine, but it's expensive. Anyone have a better option? |
| `DPO` 1-rev. | I have a caffeine addiction and I need a supplement or vitamin that has caffeine in it to take a couple times a week so that I don't have a headache all day. Any recommendations? |
| `DPO` 0-rev. | I have a caffeine addiction and I need a supplement or vitamin that has caffeine in it to take a couple times a week so that I don't have a headache all day. Any recommendations? |

### C.2.2  XSUM

| Article | *A paper detailing its proposals focuses on the need to avoid a hard border. The government stresses there should be no physical infrastructure, such as customs posts, at the border, which has almost 300 crossing points. Critics say the proposals lack credible detail, with Labour deriding the plans for the border as "a fantasy frontier". The government's paper does not envisage CCTV cameras or number plate recognition technology at the border, or set back from it. Instead, the government is arguing for a wide-ranging exemption under which small and medium-sized businesses will not have to comply with any new customs tariffs. Northern Ireland is the only part of the UK that will share a land border with an EU state post-Brexit. The future management of that border is a highly sensitive issue and is one of three main priorities in UK-EU Brexit negotiations. . . . What is the customs union? Countries in the customs union do not impose tariffs - taxes on imports - on each other's goods. Every country inside the union levies the same tariffs on imports from abroad. So, for example, a 10% tariff is imposed on some cars imported from outside the customs union, while 7.5% is imposed on roasted coffee. Other goods - such as soap or slate - have no tariffs. The UK has said it is leaving the EU's customs union because as a member it is unable to strike trade deals with other countries.* |
|---|---|
| GOLD | The government has said there must be an "unprecedented solution" for the border between Northern Ireland and the Republic after Brexit. |
| SRPO 1-rev. | The UK wants to avoid a hard border between Northern Ireland and the Republic of Ireland after Brexit by proposing two customs models to the EU. The UK also dismisses the idea of a customs border in the Irish Sea. |
| SRPO 0-rev. | Brexit Secretary David Davis wants a limited transition period to implement new customs arrangements, including consideration of the "unique circumstances" of Northern Ireland and the Republic of Ireland. The UK is trying to prevent the need for physical customs posts along the Irish border. |
| IPO 1-rev. | Theresa May wants a 'seamless' border between Northern Ireland and the Republic of Ireland after Brexit - but the EU may not accept that as viable. The UK government has published a position paper setting out its ideas on the issue. |
| IPO 0-rev. | Theresa May wants a 'seamless' border between the Republic of Ireland and Northern Ireland after Brexit - but the EU may not accept that as viable. The UK government has published a position paper setting out its ideas on the issue. |
| DPO 1-rev. | the UK is proposing a "new customs partnership" or a "highly streamlined customs arrangement" to prevent a hard border between Northern Ireland and the Republic of Ireland, in the event of no deal being reached over Brexit. |
| DPO 0-rev. | the UK is proposing a "new customs partnership" or a "highly streamlined customs arrangement" to prevent a hard border between Northern Ireland and the Republic of Ireland, in the event of no deal being reached over Brexit. |

| Article | *It follows a row over the removal of personal items from graves in Torfaen which were sent to rubbish tips. Nearly 2,000 people signed a petition calling for legislation on the issue. Public Services Minister Leighton Andrews said he was sympathetic but believed it was a matter for each local authority to "develop and justify their own approaches". Torfaen council said in June 2014 it had organised the clear-up because the over-personalisation of graves in some cemeteries had prompted complaints and made maintenance difficult. The authority apologised for any upset, but said it had made efforts to inform people about the clear-up and had allowed them two months to collect any items they wanted to keep.* |
|---|---|
| GOLD | Calls for a Wales-wide law stating what tributes can be placed on children's graves have been rejected by ministers. |
| `SRPO` {1,..,5}-rev. | Public Services Minister Leighton Andrews has rejected a petition calling for legislation on the removal of personal items from graves, saying it was a matter for each local authority to "develop and justify their own approaches". |
| `SRPO` 0-rev. | Public Services Minister Leighton Andrews has rejected a petition calling for legislation on the removal of personal items from graves. |
| `IPO` {0,..,5}-rev. | Vicky Pryce wanted revenge on ex-MP Chris Huhne over him getting points on his licence, so she took the speeding points for him in 2003, a court heard. |
| `DPO` {0,..,5}-rev. | Vicky Pryce told court she signed speeding points form for her husband Chris Huhne in revenge for him threatening their marriage over his speeding points. |

### C.3 EVALUATION DETAILS: CALCULATING WIN-RATE

Calculating the win-rate of AI models, especially in the context of comparisons against human performance or other models, involves several parameters and methodologies. In particular to calculate the win-rate of AI model vs. human, we follow these steps:

1. We run the trained model on the same input data as the test set of supervised dataset and we generate multiple completions.

2. We compare the completions using `gpt-4` with the corresponding ground truth completions of supervised dataset by assigning a point system based on outcomes (e.g., 1 for win, 0.5 for draw, 0 for loss).

3. We aggregate scores across rounds for each model and calculate the win rate.

In particular we use the following formula to calculate the win rate:

$$\text{Win Rate} = \frac{\text{Number of Wins by Model}}{\text{Total Rounds}}$$

We also report confidence intervals to indicate statistical reliability.

## D EXPERIMENTS ON ULTRA FEEDBACK

**Setup.** In our experiments, we consider the offline direct preference optimization setup to learn from human preferences (Rafailov et al., 2023). We empirically test the effectiveness of `SRPO` against two prominent offline preference learning methods, `DPO` (Rafailov et al., 2023) and `IPO` (Azar et al., 2023) as baselines. We also consider two popular extensions of `DPO`, namely, `SIMIPO` (Meng et al., 2024) and `RPO` (Chen et al., 2024).

**Implementation details.** In the case of `SRPO` and all baselines we optimize the regularization coefficient $\beta$ and the learning rate by hyper-parameter sweep over a range of possible parameters. (In the case of `RPO` we also sweep over the cross-entropy coefficient.) Also following on the recent line of work that shows using average log-likelihood is advantageous to using sum log-likelihood

in domains with varying completion length we use the average log-likelihood variant of `IPO`, `DPO`, `RPO` and `SRPO` (Grinsztajn et al., 2024; Yuan et al., 2023).

**Datasets.** We use the ShareGPT-Vicuna dataset[6] for SFT, a filtered version of the original dataset containing 53k prompt and completion pairs. For our preference training, we use the binarized version of the UltraFeedback (Cui et al., 2023)[7], which contains 64k pairwise preference data generated by various AI chatbots. For the win-rate evaluation, we use the Arena-Hard dataset[8], which contains 500 prompts, to test against `LLaMA-3.1-8B Instruct`(Dubey & others, 2024), an official post-trained version of `LLaMA-3.1-8B Base` model by the Meta LLaMA team. We generated completions using `LLaMA-3.1-8B Instruct`(Dubey & others, 2024) and Arena-Hard prompts, which we later used to compute win-rates against.

**Model Setup.** We use `LLaMA-3.1-8B Base` as base model (Dubey & others, 2024) and a Google cloud `v5litepod-256` TPUs to conduct all `LLaMA`-based training and evaluation. We first SFT the `LLaMA-3.1-8B Base` on the ShareGPT dataset. Then, we used the SFT checkpoint to initialize all preference training experiments. Below are details on the recipe for the SFT and preference training stages.

**Supervised-fine Tuning.** In the SFT stage, we train for a single epoch, using the Adam optimizer (Loshchilov & Hutter, 2019), with $\beta_1 = 0.9$ and $\beta_2 = 0.999$. We use a cosine decay learning rate scheduler(Loshchilov & Hutter, 2017) with a peak value of $2.5 \times 10^{-5}$, and decay it down to $1.25 \times 10^{-5}$. We use 10 warm-up steps and an effective batch-size of 64.

**Preference Training.** For a fair comparison across different methods, we initialize all models using the same SFT checkpoint and train them for 1 epoch using the same Adam optimizer setting: we used a learning rate of $1.25 \times 10^{-6}$, with a cosine learning rate scheduler to decay it down to $1.25 \times 10^{-7}$, with 128 warm-up steps, and an effective batch-size of 32. $\beta_1$ and $\beta_2$ values of the Adam optimizer are the same as the ones used for the SFT stage. We used the following values of $\beta$: $\beta_{\text{SRPO}} = 1.3$, $\beta_{\text{IPO}} = 1.0$, $\beta_{\text{DPO}} = 10.0$, $\beta_{\text{RPO}} = 10.0$, and $\beta_{\text{SIMIPO}} = 50.0$. For RPO, we used $1.0 \times 10^{-5}$ as the weight of cross entropy loss on the preferred completions, which is then scaled by $\beta$.

To fine-tune the models, we use a training pipeline based on `OPTAX` and `FLAX` libraries of `JAX`.[9]

**Evaluation.** We use win rates as computed by `gpt-4o` (OpenAI, 2023), as the main means for evaluation (See Sec. C.3 for more details).

**Arena-Hard Results.** We test our models using the prompts from the Arena-Hard dataset in Fig 5. For every model, we generate 0-rev. and then use these generations to revise our completions recursively from 1-rev. to 5-rev. using the self-improvement model, and measure the models' win rate against the generations of `LLaMA-3.1-8B-Instruct` model for the same prompt.

We observe that srpo 1-rev. produces the highest-quality completions, outperforming both baseline methods and other revisions of `SRPO` in terms of win rates by a significant margin. Notably, we also observe a significant drop in the win rates for all models after just one revision.

Additionally, we present SRPO's head-to-head win rate results against `DPO`, `IPO`, `RPO`, and `SIMIPO` at 0-rev. and 1-rev. We see that SRPO outperforms all other methods at both 0-rev. and 1-rev.

Table 1: ArenaHard head2head results - SRPO's WR

| vs. | DPO 0-rev. | IPO 0-rev. | RPO 0-rev. | SIMPO 0-rev. |
|---|---|---|---|---|
| SRPO 0-rev. | 53.67% | 51.10% | 51.30% | 63.1% |
| vs. | DPO 1-rev. | IPO 1-rev. | RPO 1-rev. | SIMPO 1-rev. |
| SRPO 1-rev. | 67.50% | 57.50% | 58.30% | 74.90% |

---

[6]https://huggingface.co/datasets/anon8231489123/ShareGPT_Vicuna_unfiltered
[7]https://huggingface.co/datasets/HuggingFaceH4/ultrafeedback_binarized
[8]https://lmsys.org/blog/2024-04-19-arena-hard/
[9]https://github.com/jax-ml/jax

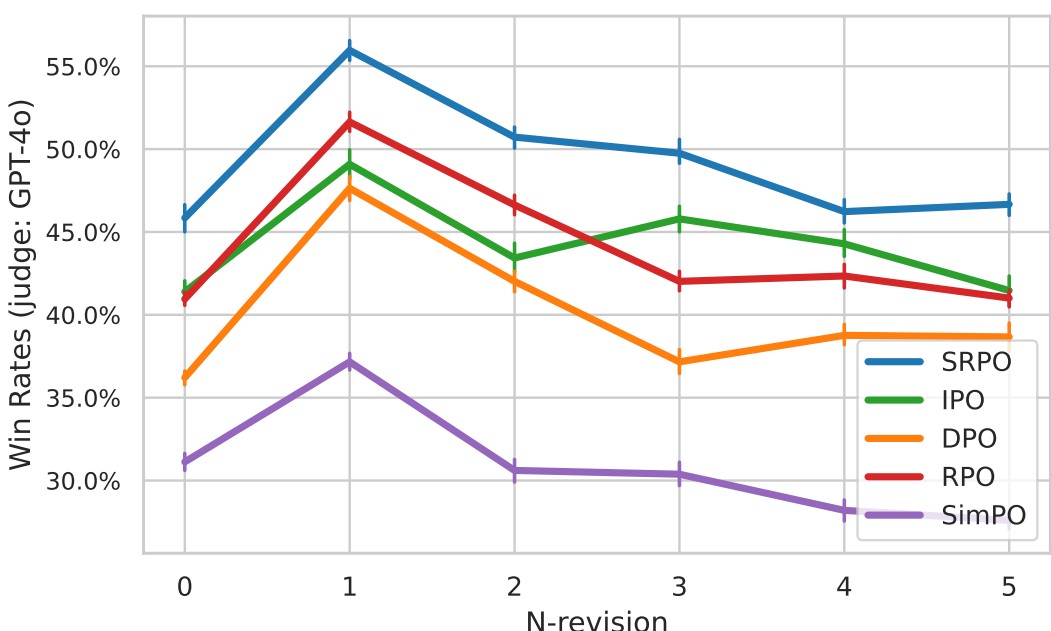

Figure 5: We present the win rates of `SRPO, IPO, DPO, SIMIPO` and `RPO` against `LLaMA-3.1-8B-Instruct` as a function of $N$-revisions for Arena-hard prompts setting. The curves represent the mean win rates, with shaded areas indicating the st.dev. across 20 bootstrap evaluations. Notably `SRPO` dominates the win-rates of all other methods across all revision steps.

