# OpenReview forum: "Self-Improving Robust Preference Optimization"
_ICLR.cc/2025/Conference — ICLR 2025 Poster_

### Official Review · Reviewer_LbBR · 2024-10-16

**Soundness:** 3
**Presentation:** 2
**Contribution:** 3
**Rating:** 6
**Confidence:** 2

**Summary:**

The authors propose Self-Improving Robust Preference Optimization (SRPO) to address two of the main drawbacks of existing RLHF methods. That is, (a) models trained with RLHF can learn from mistakes or negative examples through RL mechanism or contrastive loss at the time of training. However at the time of inference they are not equipped with an innate mechanism to correct mistakes by self-improvement. (b) The optimal solution of existing methods is highly task-dependent and thus it is difficult for them to generalize to new tasks. SRPO overcomes these shortcomings, and its effectiveness is shown also empirically.

**Strengths:**

Although this is not my field of research, the paper was sufficiently clear. Its motivations are well-stated, and the results easy to grasp. They seem relevant to the RLHF literature.

**Weaknesses:**

Probably due to my unfamiliarity with this field, I am not able to find major weaknesses to the paper. Some minor considerations follow.

Line 62: "at the time of inference It would be very" should be changed to "at the time of inference, it would be very".

What is $\mathbb{R}^*_+$ in line 162?

Line 195: "Eq. equation 2" should be "equation 2". This typo is repeated throughout the manuscript.

**Questions:**

Is the problem studied by the authors related to continual learning under specific trade-offs [1]?

---

[1] https://arxiv.org/abs/2305.14782

---

> ### Author Response · Authors · 2024-11-23
> **Response to Reviewer Comments/Questions**
>
> We thank the reviewer for their valuable comments and insightful questions. We have addressed the concerns and queries raised by the reviewer in this response.
>
> **Questions:**
>
> **What is  $R^+_∗$ in line 162?** It is the set of strictly positive real numbers.
>
> **Relation to Continual Learning:** We thank the reviewer for highlighting the potential connection between our work and continual learning (CL), particularly in relation to the specific work cited. While there may be conceptual overlaps, we believe that our study is better framed within the context of continual self-improvement rather than traditional continual learning. The primary distinction lies in the approach we take: the variant of SRPO presented in this paper trains the self-improvement policy in a single, static training phase rather than iteratively adapting to new tasks or data distributions over time, as is typical in CL.
>
> Continual learning often focuses on managing trade-offs such as overcoming catastrophic forgetting while adapting to new tasks or data. In contrast, our work centers on developing a robust self-improvement policy that enhances model outputs iteratively but within a fixed training framework. That said, extending SRPO to the continual learning setting, where the self-improvement policy evolves as new data or tasks are introduced, could be a promising direction for future research. Such an extension could explore dynamic adaptation of the self-improvement policy, potentially integrating ideas from continual learning to balance adaptation and retention in long-term iterative improvement processes.

---

> > ### Comment · Reviewer_LbBR · 2024-11-26
> > **Thank you**
> >
> > I'm happy with the author response, and I'm happy to keep my score

---

### Official Review · Reviewer_hrBf · 2024-10-27

**Soundness:** 4
**Presentation:** 3
**Contribution:** 3
**Rating:** 6
**Confidence:** 3

**Summary:**

This paper proposes Self Improving Robust Preference Optimization (SRPO), a new method of LLM fine tuning. The authors propose using in-context learning to iteratively generate higher quality completions while being robust to the underlying offline dataset. The paper proposes a min-max objective which aims to maximize the quality of the improvement (of the improvement model) and minimize the amount by which you can be improved (for another language model). The paper then analyzes optimizations for each of the models independently and derive a convex combination of these losses. Finally, the authors experimentally verify their theory with experiments on TL;DR and XSum.

**Strengths:**

The observations made by this paper are well supported empirically. In particular, they use a nice synthetic bandit example to show robustness. Further, they experimentally verify on the TL;DR dataset and XSum on out of distribution examples. This paper further does a good job elucidating the derivation of the objective.

**Weaknesses:**

- The proposed method empirically performs similar to other methods under the same amount of inference compute.
- The paper does not provide much theoretical justification of the combination loss.

**Questions:**

- DPO and IPO both have "revisions"? Does this mean they received the same in context prompt as SRPO?
- The primary thesis of this work is that current preference optimization methods should not expect the ideal completion to be written and PPO/DPO/IPO are solving a fundamentally different task. In that vein, how does PPO compare under these same conditions? Is the lack of robustness also observed?
- Overoptimization has been observed in DPO and other direct alignment algorithms. Is this observed with SRPO as well? Why/Why not? [0]


[0] https://arxiv.org/abs/2406.02900

---

> ### Author Response · Authors · 2024-11-23
> **Response to Reviewer Comments/Questions**
>
> We thank the reviewer for their valuable comments and insightful questions. We have addressed the concerns and queries raised by the reviewer in this response, as well as in the *revised manuscript submitted alongside it*. Additionally, we are in the process of preparing *another revised manuscript*, which will include new experimental results specifically aimed at addressing the reviewer’s concerns regarding the experiments.
>
> **Weaknesses**
>
> **Similarity in Performance with baselines:** It is accurate that, in the case of zero revisions, our method performs comparably to IPO and still outperforms DPO in the case of TLDR. (Though our **new results on the challenging Arena Hard** and our prior results on XSUM shows that SRPO outperforms both IPO and DPO w/o revision.) However, the true strength of SRPO becomes evident after multiple revisions, where it consistently achieves superior performance compared to IPO when subjected to the same number of revision cycles. This demonstrates that SRPO utilizes the same amount of compute more effectively, leveraging its iterative refinement process to deliver better outcomes than its competitors. By optimizing each revision step, SRPO ensures that the improvements compound over iterations, highlighting its efficiency in achieving high-quality results over multiple refinement steps.
>
> **Theoretical Justification for Combination Loss:** We will formalize this theoretical result in Theorem 1, which will be thoroughly detailed in Sections 4.1 (Main result), 4.2 (Proof) of the revised submission (to be included with this rebuttal). This result serves to justify the use of the combination loss in our framework. Specifically, Section 4.2.3 will be dedicated to presenting a rigorous proof of the combination loss, providing a clear and comprehensive explanation of its mathematical foundations.
>
> **Questions**
>
> **IPO/DPO multi-step prompt:** Yes, for consistency and fairness in evaluation, we provide the models trained by IPO and DPO with the same in-context prompt used for SRPO. This ensures that all models are tasked with improving their outputs under identical conditions, allowing for a direct and meaningful comparison of their ability to refine and enhance completions. By using the same prompt structure, we aim to isolate the differences in performance attributable to the underlying training and optimization methods, rather than discrepancies in the prompts or contextual inputs.
>
>
> **Robustness of PPO:** PPO and DPO optimize the same underlying objective, and as a result, they are both subject to similar limitations in terms of robustness. Specifically, their performance can be sensitive to the behavior policy $\mu$, which governs the data distribution used for training. This dependency can lead to a lack of robustness, as noted in prior work (e.g., Section 3.3 of Munos et al., 2023). While PPO is often used in online settings where the behavior policy can be dynamically adjusted, this sensitivity still poses challenges when transferring to offline or fixed-policy scenarios.
>
> In this submission, we focus primarily on offline methods, as our goal is to address the challenges associated with static datasets. Consequently, we have chosen to compare SRPO against DPO, which is directly relevant to our context. However, extending the evaluation to include PPO under comparable conditions is an interesting direction for future work.
>
> **Over-optimization in SRPO:** Yes, over-optimization is observed in SRPO as well. Similar to other direct alignment methods, SRPO experiences a decline in performance, measured by win-rate, after approximately 3-4 epochs of training. This phenomenon occurs because prolonged training can cause the model to overfit to the specific patterns in the training data or over-optimize for the alignment loss, potentially at the expense of generalization. In essence, the model becomes overly tailored to the preference data it has seen, which may not fully represent the diversity or complexity of real-world tasks.
>
> To mitigate this issue, careful strategies such as early stopping, regularization techniques, or validation-based checkpoints could be employed to prevent over-optimization during training. Additionally, exploring ways to improve the diversity and representativeness of the training data, or leveraging ensemble methods, may help SRPO maintain robust performance over extended training epochs. Further analysis and experimentation are warranted to understand the nuances of over-optimization in SRPO and develop strategies to address it effectively.
>
> **Munos et al 2023:**  *Nash Learning from Human Feedback*

---

> ### Author Response · Authors · 2024-11-27
> **New Experiments Highlighting the Superiority of SRPO over Other Preference Optimization Methods**
>
> In response to the reviewers' concerns regarding **the similarity in performance between SRPO** and other baseline methods, when using same amount of compute, we would like to direct attention to our latest results. Specifically, on the challenging **Arena Hard** benchmark, SRPO demonstrates consistent and significant improvements over a range of prominent preference optimization methods, even without additional revisions (i.e., no significant extra compute needed at inference time to achieve better performance).
> A snippet of these results is presented in the following table (Full report of results is included in Appendix D of revised manuscript):
>
> **Win-rates against  Llama 3.1 8B Instruct (Judged by GPT 4)**
>
>
> | Method | 0-revision | 1-revision |2-revision|
> |:-----------|:------------:|:------------:|:------------:|
> | **SRPO**|$\mathbf{45.9\pm1.6}$\%|$\mathbf{56.0\pm1.1}$\%|$\mathbf{50.7\pm1.2}$\%|
> | IPO    | $41.4\pm1.3$\%| $49.1\pm1.6$\%|$43.4\pm1.7$\%|
> | DPO   | $36.2\pm0.9$\%| $ 47.6\pm1.5$\%|$42.0\pm1.2$\%|
> | RPO   | $41\pm0.7$\%| $51.6\pm1.1$\%|$46.6\pm1.2$\%|
> | SIMPO   | $31.1\pm1$\%| $ 37.1\pm1$\%|$30.6\pm1.4$\%|
>
>
> This evidence provides a compelling demonstration of SRPO's superiority over prior approaches, especially in, Arena hard arguably one of the most challenging benchmarks in generative AI.

---

> > ### Comment · Reviewer_hrBf · 2024-11-28
> > **response to authors**
> >
> > Thank you for the response to my questions and concerns. I think that the authors have significantly increased the quality of this manuscript. The area hard SRPO results are interesting, both because this gives greater insight into the true performance of the method, and because it does not actually seem to benefit significantly from revisions. I think that the current quality of the manuscript is acceptable, and I will increase my score **with the understanding that the authors look more into the impact of revisions** which I think can be improved. However, I believe that this method is an improvement and the community would benefit from this work.

---

### Official Review · Reviewer_Pvzi · 2024-11-03

**Soundness:** 3
**Presentation:** 3
**Contribution:** 2
**Rating:** 6
**Confidence:** 4

**Summary:**

This paper introduces a novel method, Self-Improving Robust Preference Optimization (SRPO), which aims to address two primary limitations in current preference optimization techniques. First, existing methods generally lack an inherent self-improvement mechanism at inference time, which limits adaptability. Second, they often rely heavily on the training task and distribution used to generate preference data, which affects the robustness of solutions. To address these issues, SRPO is designed as an offline preference optimization method that leverages a min-max formulation to learn a robust generative policy. This policy generates completions that require minimal adjustment when optimized through a self-improvement policy. The authors apply a DPO/IPO-inspired derivation to show that this min-max objective can be effectively optimized through standard supervised learning techniques. Experimental results on the TL;DR Summarization dataset indicate that SRPO achieves better in-distribution (ID) and out-of-distribution (OOD) performance compared to DPO and IPO baselines.

**Strengths:**

The paper is clearly written and easy to follow, with each step of the SRPO algorithm systematically derived.

The derivations provided in the paper appear rigorous, demonstrating a well-founded approach to preference optimization. The use of a min-max objective with a focus on self-improvement mechanisms is an interesting contribution.

**Weaknesses:**

The evaluation is currently limited to a single dataset (TL;DR Summarization) and only compares SRPO against two baselines, DPO and IPO. Conducting experiments on additional datasets would strengthen the empirical claims related to robustness.

The paper lacks (empirical) comparisons to more recent preference optimization methods, such as SimPO (Meng et al., 2024) and RPO (Liu et al., 2024), which integrate recent advancements over DPO.

In Figure 3, SRPO achieves improved performance with revisions, likely due to its self-improvement mechanism. However, unlike DPO and IPO, which lack this self-correction feature, SRPO appears to incur additional inference costs to achieve its performance benefits.

(Meng et al., 2024) SimPO: Simple Preference Optimization with a Reference-Free Reward

(Liu et al., 2024) Provably Mitigating Overoptimization in RLHF: Your SFT Loss is Implicitly an Adversarial Regularizer

**Questions:**

Please see above.

---

> ### Author Response · Authors · 2024-11-23
> **Response to Reviewer Comments/Questions**
>
> We thank the reviewer for their valuable comments. We have addressed the concerns and queries raised by the reviewer in this response, as well as in the *revised manuscript submitted alongside it*. Additionally, we are in the process of preparing *another revised manuscript*, which will include new experimental results specifically aimed at addressing the reviewer’s concerns regarding the experiments.
>
> **Weaknesses:**
>
> **The paper lacks multiple empirical comparisons:** We are currently evaluating SRPO on  the **ultra-feedback** dataset, which has been designed to provide deeper insights into the model's ability to align with complex human preferences. The ultra-feedback dataset allows for a more rigorous testing of SRPO's performance across diverse scenarios, helping to assess its scalability and effectiveness in handling nuanced feedback. We will present the results of these evaluations, including detailed analysis and comparisons, in a revised draft that will be submitted *before the rebuttal deadline.*
>
> **Comparison with SimPO and RPO:** We are in the process of launching experiments for SimPO and RPO to thoroughly evaluate their performance in comparison to SRPO. The results of these experiments, along with a detailed comparison between SimPO, RPO, and SRPO, will be included in the revised version of the manuscript, which will be submitted prior to the rebuttal deadline. Additionally, we will expand the discussion in the related work section to provide a comprehensive comparison of SimPO and RPO, highlighting their methodologies, strengths, and limitations relative to SRPO. This will ensure that the context and significance of our contributions are clearly articulated.
>
> **Significance of improvement:** We acknowledge that SRPO involves a trade-off between achieving higher-quality completions and the additional computational overhead required to compute these improved completions. However, we believe this trade-off is justified when the additional compute results in significantly better performance. Utilizing extra computation enables us to achieve high-quality outputs even with smaller models, offering several practical advantages. For instance, this approach facilitates the deployment of high-performance language models on edge devices or resource-constrained environments without demanding excessive memory or computational resources. By optimizing the balance between computational cost and model performance, SRPO provides a pathway for creating efficient, scalable, and portable language solutions that meet diverse application needs.

---

> ### Author Response · Authors · 2024-11-26
> **New Experiments: SRPO vs. DPO and IPO on Ultra Feedback Dataset  **RESULTS UPDATED****
>
> In response to the reviewer request for more experiments, we conducted additional experiments to evaluate the self-improvement capabilities of SRPO by post-training on a Llama 3.1 8B base model using Ultra Feedback dataset (Cui et al 2024), and comparing its performance against IPO and DPO. Below is a snippet of the results (the full results is available in the Appendix D  of the revised manuscript) using the [Arena-Hard prompts](https://lmsys.org/blog/2024-04-19-arena-hard/).
>
> **Win-rates against  Llama 3.1 8B Instruct (Judged by GPT 4)**
>
> | Method | 0-revision | 1-revision |2-revision|
> |:-----------|:------------:|:------------:|:------------:|
> | SRPO|$\mathbf{40.5\pm1.7}$\%|$\mathbf{54.2\pm1.6}$\%|$\mathbf{47.8\pm1.6}$\%|
> | IPO    | $\mathbf{41.4\pm1.3}$\%| $49.1\pm1.6$\%|$43.4\pm1.7$\%|
> | DPO   | $36.2\pm0.9$\%| $ 47.6\pm1.5$\%|$42.0\pm1.2$\%|
>
> The new findings further reinforce our previous observations: SRPO significantly outperforms both DPO and IPO after one revision. Additionally, SRPO demonstrates competitive performance with IPO (within margins of error) in the zero-revision scenario, while also outperforming DPO at this stage.
>
> **Cui et al 2024** UltraFeedback: Boosting Language Models with High-quality Feedback
>
> **-------------------------------------------------------------------------------Important Result Update------------------------------------------------------------**
>
> We conducted a more comprehensive evaluation of our results by calculating the AI win rates across all checkpoints sweep trained with SRPO, IPO, and DPO (due to AI API quota limit we could not do this exhaustive hyper sweep before).
>
> The analysis revealed that, if it is tuned carefully, SRPO consistently outperforms all baselines by a significant margin across all revisions. In contrast, similar exhaustive evaluations and hyper-parameter tuning  for IPO and DPO did not show any notable improvements in comparison to previously reported results. A snippet of updated win-rates on Arena Hard are presented below (complete report of our result will be included in the revised manuscript).
>
> **Win-rates against  Llama 3.1 8B Instruct (Judged by GPT 4--- UPDATED---)**
>
> | Method | 0-revision | 1-revision |2-revision|
> |:-----------|:------------:|:------------:|:------------:|
> | **SRPO (updated)**|$\mathbf{45.9\pm1.6}$\%|$\mathbf{56.0\pm1.1}$\%|$\mathbf{50.7\pm1.2}$\%|
> | IPO    | $41.4\pm1.3$\%| $49.1\pm1.6$\%|$43.4\pm1.7$\%|
> | DPO   | $36.2\pm0.9$\%| $ 47.6\pm1.5$\%|$42.0\pm1.2$\%|

---

> ### Author Response · Authors · 2024-11-27
> **New Experiments: SRPO vs. SIMPO and RPO**
>
> In response to the reviewers' request for a **comparison between SRPO, SIMPO, and RPO**, we provide a snippet of win-rate results in the table below trained on **Ultra-feedback** dataset  and evaluated on the challenging **Arena-Hard** prompts. (Full report of results is included in Appendix D of revised manuscript.)
>
> **Win-rates against Llama 3.1 8B Instruct (Judged by GPT 4)**
>
> | Method | 0-revision | 1-revision |2-revision|
> |:-----------|:------------:|:------------:|:------------:|
> | **SRPO**|$\mathbf{45.9\pm1.6}$%|\$\mathbf{56.0\pm1.1}$%|$\mathbf{50.7\pm1.2}$%|
> | RPO   | $41\pm0.7$\%| $51.6\pm1.1$\%|$46.6\pm1.2$\%|
> | SIMPO   | $31.1\pm1$\%| $ 37.1\pm1$\%|$30.6\pm1.4$\%|
>
> Our analysis indicates that SRPO outperforms both SIMPO and RPO by a significant margin across all metrics. Notably, this superior performance is achieved despite SRPO not utilizing the additional cross-entropy regularization term present in RPO, which is known to enhance performance of RPO in comparison with vanila DPO.

---

> > ### Comment · Reviewer_Pvzi · 2024-12-02
> >
> > I thank the authors for additional experiments (which strengthen the paper) and for clarifying my concerns. I will raise my score.

---

### Official Review · Reviewer_Nqte · 2024-11-05

**Soundness:** 3
**Presentation:** 3
**Contribution:** 3
**Rating:** 6
**Confidence:** 3

**Summary:**

This paper studies the problem of reinforcement learning from human feedback (RLHF). It designs a novel RLHF algorithm called SRPO based on training a robust self-improving policy by solving a min-max problem. It explains the math in detail on how to find the solution to this min-max problem. The experiment results show that SRPO achieves a higher winning rate against the gold dataset than existing state-of-the-art RLHF algorithms like DPO and IPO.

**Strengths:**

1. This paper studies the critical problem of finding a more efficient RLHF algorithm. It could have great potential in many important real-world applications.

2. The idea of using a self-improving policy in solving the RLHF problem is novel.

3. The mathematical explanation of how to solve the min-max learning problem they propose is thorough.

3. The SRPO algorithm aims at training a robust policy against the quality of the dataset, making it more trustworthy than many other RLHF algorithms.

4. The experiment design is reasonable in general, and the results look positive.

**Weaknesses:**

1. There is no theoretical guarantee of the learning outcome. This makes the whole theoretical part weak. Is there a chance to provide any theoretical guarantee on the performance of the policy learned by SRPO under some assumptions?

2. The design of SRPO algorithm is novel, but the description of its motivation can be improved. Currently, the motivation is that 'Instead, it is more natural to learn that given a query x and a completion y what would be the improved completion upon y'. However, in general, you can also say that to some other learning algorithms as long as they improve the quality of their policy to generate better completion after each training iteration. It is better to find some more unique motivation for the design of the SRPO algorithm.

3. The SRPO algorithm's winning rate against other algorithms is not provided. This paper only measures the winning rate against a gold-standard dataset. A policy that has a higher winning rate against a fixed dataset than that of another policy does not necessarily mean this policy is better than that policy. To verify that the policy learned by SRPO is better than the policies learned by other algorithms, it is necessary to compare these policies against each other directly.

**Questions:**

1. It can be beneficial to provide more explanation of the self-improving policy. This concept is not common in many RLHF literatures. Readers may be curious about how these policies work and how they are implemented. Providing more details can make this paper easier to follow.

2. Can you provide more details about the evaluation? For example, what are the parameters, such as the accuracy compared to humans, of the AI evaluation platform? Exactly how do we measure the winning rate between any two models? Revealing such details can make this work much easier to reproduce.

---

> ### Author Response · Authors · 2024-11-23
> **Response to Reviewer Comments/Questions**
>
> We thank the reviewer for their valuable comments and insightful questions. We have addressed the concerns and queries raised by the reviewer in this response, as well as in the *revised manuscript submitted alongside it*. Additionally, we are in the process of preparing *another revised manuscript*, which will include new experimental results specifically aimed at addressing the reviewer’s concerns regarding the experiments.
>
> **Weaknesses:**
> 1. Our derivations in Section 4 establish a theoretical foundation for the convergence of SRPO to the minimax solution of the objective defined in Equation 1. In the revised submission (to be included with this rebuttal), we will formalize this result as Theorem 1, detailed in Sections 4.1 (Main result) , 4.2 (Proof). This theorem demonstrates that the complex optimization objective of SRPO can be reformulated as a standard supervised learning problem, which is scalable and solvable using an offline human preference dataset.
>
> 2. In the revised submission (to be included with this rebuttal), we have refined the motivation in Section 2 to more effectively highlight the importance of learning self-improvement policies and provide a detailed explanation of why this is critical for training GPT-style models. Specifically, we address the issue of exposure bias in language models, a challenge that arises because these models generate tokens sequentially based on their prior outputs. This sequential generation can lead to error accumulation when an incorrect token is produced, propagating mistakes throughout the output. We emphasize that self-refinement models offer a promising solution to this problem by enabling models to correct their own outputs. However, training such models through supervised fine-tuning (SFT) requires constructing datasets with both initial completions and their corrected versions, a process that is resource-intensive and susceptible to overfitting to specific patterns or neglecting the refinement process altogether. To address these limitations, we introduce SRPO as an alternative approach based on direct preference optimization. SRPO teaches models to align less-preferred outputs with more-preferred ones, focusing on the mechanics of improvement rather than simply predicting the most-preferred completion. This approach provides a more scalable and natural framework for learning self-improvement policies.
>
> 3. We are in the process of preparing these evaluations and will include head-to-head comparisons in a revised submission prior to the rebuttal deadline.
>
> **Questions:**
> 1. In Section 2 of the revised manuscript (to be submitted with this rebuttal), we will provide a more in-depth explanation of the significance of self-improvement policies. Specifically, we will emphasize how self-improvement can effectively address the issue of exposure bias in language models, a critical challenge where error accumulation arises due to sequential token generation. Additionally, we will highlight the necessity of advancing existing methods for training self-improvement policies to enhance their effectiveness and scalability.
>
> 2. We will include detailed information on computing the win rate against humans in Appendix C.3 of the revised manuscript (to be submitted with this rebuttal). Regarding the accuracy of AI evaluators compared to humans, we refer the reviewer to prior work, such as Arawjo et al. (2023), which establishes the reliability and accuracy of AI-based evaluators.
>
>  **Arawjo et al. (2023):** *Who Validates the Validators? Aligning LLM-Assisted Evaluation of  LLM Outputs with Human Preferences.*

---

### Meta-Review · Area_Chair_N5DS · 2024-12-31

**Metareview:**

This work proposes a new preference optimization algorithm, self-improving robust preference optimization (SIRPO).  This DPO variant is capable of performing in-context learning to self-improve during execution of the policy.  Importantly, the authors also propose a robust training method which improves out-of-distribution generalization, a problem of considerable interest to the community.

Reviewers generally found the paper to be well-written and well-motivated.  Initially there were some concerns about the generalizability of the results and theoretical underpinnings, but this was addressed during the rebuttal period, in which the authors added an additional benchmark task (post-training llama 3.1 using the Ultra Feedback dataset, and evaluating on Arena-Hard prompts).  SRPO was evaluated by comparing win rates against RPO and SIMPO baselines, and appears to robustly outperform these methods under a variety of conditions and hyper parameter configurations.  Theoretical justifications of the loss function were addressed via Theorem 1 and revisions.

Overall, reviewers and the AC found this work compelling enough for publication, and I commend the authors on their compelling rebuttal and revision.  Please work to thoroughly address any remaining issues and analyses suggested by the reviewers.  For the CR, I would recommend not including the first figure (especially on the first page, since the reader will have no context on how to interpret it!).  The other variants of this figure are more informative, and an illustration is not needed to make the rather intuitive idea behind this work.

**Additional Comments On Reviewer Discussion:**

The reviews were generally positive, but some questions around the generalizability, evaluation methodology, and theoretical basis were raised.  Through revisions and additional experiments, the authors compellingly addressed reviewer concerns, resulting in all reviewers advocating for (weak) accept rating.  While background knowledge was mixed among reviewers, some reviewers had sufficient expertise and detailed comments for me to feel comfortable with accepting this work.

---

### Decision · Program_Chairs · 2025-01-22

Accept (Poster)